# Catfish Egg Lectin Enhances the Cytotoxicity of Sunitinib on Gb3-Expressing Renal Cancer Cells

**DOI:** 10.3390/biomedicines11082317

**Published:** 2023-08-21

**Authors:** Jun Ito, Shigeki Sugawara, Takeo Tatsuta, Masahiro Hosono, Makoto Sato

**Affiliations:** 1Department of Urology, Faculty of Medicine, Tohoku Medical and Pharmaceutical University, 1-15-1 Fukumuro, Miyagino-ku, Sendai 983-8536, Japan; ms.hifu@tohoku-mpu.ac.jp; 2Division of Cell Recognition Study, Institute of Molecular Biomembrane and Glycobiology, Tohoku Medical and Pharmaceutical University, 4-4-1 Komatsushima, Aoba-ku, Sendai 981-8558, Japan; ssuga@tohoku-mpu.ac.jp (S.S.); t-takeo@tohoku-mpu.ac.jp (T.T.); mhosono@tohoku-mpu.ac.jp (M.H.)

**Keywords:** globotriaosylceramide, rhamnose-binding lectin, sunitinib, renal cell carcinoma

## Abstract

Metastatic renal cell carcinoma (RCC) is not sufficiently responsive to anticancer drugs, and thus, developing new drugs for advanced RCC remains vital. We previously reported that the treatment of globotriaosylceramide (Gb3)-expressing cells with catfish (*Silurus asotus*) egg lectin (SAL) increased the intracellular uptake of propidium iodide (PI) and sunitinib (SU). Herein, we investigated whether SAL pretreatment affects the intracellular uptake and cytotoxic effects of molecular-targeted drugs in RCC cells. We analyzed Gb3 expression in TOS1, TOS3, TOS3LN, and ACHN human RCC cells. Surface Gb3 expression was higher in TOS1 and TOS3 cells than in TOS3LN and ACHN cells. In the PI uptake assay, 41.5% of TOS1 cells and 21.1% of TOS3 cells treated with SAL were positive for PI. TOS1 cell viability decreased to 70% after treatment with 25 µM SU alone and to 48% after pretreatment with SAL (50 µg/mL). Time-series measurements of the intracellular fluorescence of SU revealed significantly enhanced SU uptake in SAL-treated TOS1 cells compared to control cells. SAL treatment did not increase PI uptake in normal renal cells. Our findings suggest that adequate cytotoxic activity may be achieved even when SU is administered at a sufficiently low dose not to cause side effects in combination with SAL.

## 1. Introduction

The number of renal masses accidentally detected by ultrasound or computed tomography has increased due to the increased use of diagnostic imaging tools in patients complaining of nonspecific abdominal or back pain. Approximately 80% of renal masses are considered to be renal cell carcinoma (RCC) [1]. While early detection of renal masses has improved the overall treatment outcome of RCC, RCC is not sufficiently responsive to anticancer drugs or radiation therapy once it becomes metastatic. Therefore, new drugs for the treatment of advanced RCC are required.

Several therapeutic drugs for treating advanced RCC have been developed to meet this demand in recent years. Molecular targeting drugs exert antitumor effects by inhibiting intracellular signaling involved in the growth of tumor cells and vascular endothelial cells and therefore are a promising option. Tyrosine kinase inhibitors (TKIs) targeting the vascular endothelial growth factor (VEGF), which acts on the vascular endothelium, inhibitors of the mammalian target of rapamycin, which is involved in tumor growth, and recently, multi-tyrosine kinase inhibitors of AXL receptor tyrosine kinase (AXL) and MET receptor tyrosine kinase (MET), which are activated by the inhibition of the VEGF, have been approved and are currently being used for the treatment of unresectable/metastatic RCC [2,3]. Despite the advent of immune checkpoint agents and other drugs with novel mechanisms of action, TKIs continue to play an important role in the treatment of renal cancer [4]. However, these molecular targeting drugs can induce multiple dose-related side effects, sometimes leading to treatment discontinuation. Therefore, new therapeutic strategies are required.

Lectins, which are proteins that can bind to glycosphingolipids (GSLs), have recently received increased research interest for their role as information devices in living organisms [5]. *Silurus asotus* egg lectin (SAL) isolated from catfish eggs has been found to bind to globotriaosylceramide (Gb3) expressed on the surfaces of cells [6]. We have previously studied the effects of SAL in Gb3-expressing tumor cells [7,8]. In a study on Gb3-expressing Raji Burkitt’s lymphoma cells, we found that the binding of SAL to Gb3 increased the intracellular uptake of propidium iodide (PI). Moreover, SAL induced cell permeability and enhanced the cytotoxicity of doxorubicin [7]. However, the molecular mechanism of PI and doxorubicin uptake has not been elucidated. Furthermore, our previous study showed that the binding of SAL to Gb3 expressed on the surfaces of HeLa cells promoted the cellular uptake of the tyrosine kinase inhibitor sunitinib (SU), which is used to treat RCC, and delayed its excretion, which was associated with a significant decrease in cell survival [8]. These findings suggest that SAL may enhance the uptake of therapeutic agents into Gb3-expressing tumor cells and reduce their optimal dose.

In this study, we evaluated whether the combined use of SU and SAL would reduce the viability of RCC cells. In addition, we investigated whether the increased uptake and delayed excretion of therapeutic agents in tumor cells, as observed in our previous study, would also occur in RCC cells. This study aimed to provide a new therapeutic approach to RCC treatment.

## 2. Materials and Methods

### 2.1. Lectin and Cell Lines 

SAL was purified using a method described previously [6]. The TOS1, TOS3, TOS3LN, and ACHN human renal cell cancer cell lines were a gift from Dr. Makoto Sato (Department of Urology, Tohoku University, School of Medicine, Sendai, Japan). Human renal proximal tubular epithelial cells (HRPTEC) were purchased from KURABO Industries (Osaka, Japan). The renal cancer cells were cultured in Dulbecco’s Modified Eagle Medium (DMEM) (Wako, Osaka, Japan) supplemented with 10% (*v*/*v*) fetal bovine serum (FBS) and antibiotic-antimycotic solution (penicillin (100 IU/mL), streptomycin (100 µg/mL), and amphotericin B (0.25 µg/mL); Life Technologies, Carlsbad, CA, USA) at 37 °C in a 95% air/5% CO_2_ atmosphere. HRPTEC were cultured in a RenaLife Comp kit (KURABO Industries) supplemented with RenaLife Life Factors (KURABO Industries) at 37 °C in a 95% air/5% CO_2_ atmosphere.

### 2.2. Flow-Cytometric Analysis of Cellular Propidium Iodide (PI) Uptake to Measure Cell Permeability

Cells (2 × 10^5^) were treated or not with 100 µL of SAL (50 µg/mL Dulbecco’s phosphate-buffered saline (D-PBS)) at 4 °C for 30 min and washed thrice with D-PBS. PI uptake was quantified using a MEBCYTO apoptosis kit (MBL, Nagoya, Japan) and a FACSCalibur flow cytometer (BD Biosciences, Franklin Lakes, NJ, USA). 

### 2.3. Cell Viability Assay

Cytotoxic activity was determined with a trypan blue (0.5% *w*/*v*) exclusion assay. Cell viability was determined using the cell counting kit-8 (CCK-8) assay (Dojindo Laboratories, Kumamoto, Japan). Cells were seeded into a 96-well flat-bottom plate at 5 × 10^3^ cells/well (90 µL) and treated with SAL (final concentration, 50 µg/mL) for 24 h. Then, CCK-8 solution (10 µL) was added to each well, and the cells were incubated at 37 °C for 4 h. The absorbance at 450 nm was measured using a Tecan Infinite F200 PRO microplate reader (Tecan Austria GmbH, Männedorf, Austria). 

### 2.4. Flow-Cytometric Analysis of Cell Surface Gb3 Expression 

Cells (2 × 10^5^) were treated or not with an anti-Gb3 monoclonal antibody (mAb) (BGR23, mouse IgG2b; Tokyo Kasei, Tokyo, Japan) diluted at a 1:500 ratio in D-PBS (100 μL) at 4 °C for 30 min and washed thrice with D-PBS. Then, they were incubated with Alexa Fluor (AF) 488-conjugated goat anti-mouse IgG (H + L) (Molecular Probes, Invitrogen AG, Basel, Switzerland) diluted at a 1:2500 ratio in D-PBS (100 µL) at 4 °C for 30 min. Gb3 expression on the cell surface was quantified using a BD FACSCalibur™ Flow Cytometer (BD Biosciences).

### 2.5. Thin-Layer Chromatography (TLC) for Glycolipid Expression Analysis 

Cells (1 × 10^6^) were suspended in a chloroform-methanol solution (2:1, *v*/*v*), incubated at 37 °C for 1 h, and then centrifuged at 1000× *g* for 10 min. The supernatant was recovered in a glass tube. The pellet was resuspended in a chloroform-methanol-water solution (1:2:0.8, *v*/*v*/*v*), incubated at 37 °C for 2 h, and centrifuged at 1000× *g* for 10 min. The supernatant was collected and evaporated to dryness under nitrogen gas. The residue was dissolved in 20 µL of chloroform-methanol (2:1, *v*/*v*), placed on a high-performance TLC plate (Merck KGaA, Darmstadt, Germany), and developed using a solvent system of chloroform-methanol-water (60:35:8, *v*/*v*/*v*). Gb3 was visualized by spraying 0.5% orcinol in 10% sulfuric acid. Neutral glycosphingolipids, including cerebrosides, lactosylceramide (LacCer), Gb3, and Gb4, were purchased from Matreya (State College, PA, USA).

### 2.6. Analysis of the Effect of a Combination of SU and SAL 

To assess the effect of SU, cells (5 × 10^3^) were treated with SU (0, 6.25, 12.5, 25, and 50 µM) at 37 °C for 24, 48, and 72 h. To determine the effect of a combination of SU and SAL, cells were first incubated with SAL (50 µg/mL) in DMEM supplemented with FBS for 24 h and then with SU (0 and 25 µM) at 37 °C for another 24 h. Cell viability was determined using the CCK-8 assay as described above. Annexin V-positive cells were detected using the MEBCYTO apoptosis kit (MBL) as mentioned above. Bright-field images were acquired using an inverted microscope (model IX71; Olympus Corporation, Osaka, Japan) with a 10× or 100× objective lens. 

### 2.7. Analysis of the Effect of a Combination of Other Molecular-Targeted Agents and SAL 

To assess the effect of other molecular-targeted agents, cells (5 × 10^3^) were treated with pazopanib (0, 3.125, 6.25, 12.5, and 25 µM), axitinib (0, 3.125, 6.25, 12.5, and 25 µM), or everolimus (0, 5, 10, 20, 40, and 80 µM) at 37 °C for 24 or 48 h. To determine the effect of a combination of these agents and SAL, cells were first incubated with SAL (50 µg/mL) in DMEM supplemented with FBS for 24 h and then with pazopanib (0 and 25 µM), axitinib (0 and 25 µM), or everolimus (0, 35, and 40 µM) at 37 °C for another 24 or 48 h. Cell viability was determined using the CCK-8 assay as described above.

### 2.8. Efflux of SU from SAL-Treated TOS1 Cells 

Cells (5 × 10^5^) were incubated in DMEM with (50 µg/mL) or without SAL at 37 °C in a 95% air/5% CO_2_ atmosphere for 24 h and then treated with SU (25 µM) for another 30 min. After SU was removed from the wells, the cells were observed at 3 or 6 h intervals for 24 h. SU efflux was visualized using an Olympus FV1000 confocal scanning microscope (Olympus).

### 2.9. Time-Series Measurements of Intracellular SU Contents

Cells were seeded in a CellCarrier™ 96-well microplate (PerkinElmer Cellular Technologies Germany GmbH, Hamburg, Germany) at a density of 1 × 10^4^ cells/well and cultured in DMEM with (50 µg/mL) or without SAL at 37 °C in a 95% air/5% CO_2_ atmosphere. The cells were stained with Hoechst 33342 (Dojindo Laboratories), and the plate was scanned using an Operetta CLS High Content Analysis System (PerkinElmer) with a 40× objective lens in the confocal mode in a pre-warmed live cell chamber set at 37 °C in a 95% air/5% CO_2_ atmosphere. Fluorescence images were captured using the Hoechst 33342 channel at 488 nm before and at 3 or 30 min intervals after adding SU (final concentration, 25 µM) for a total period of 1.5 h. Fluorescence signals were quantified using the Harmony 4.9 software (PerkinElmer). 

### 2.10. Reverse Transcription-Quantitative Real-Time Polymerase Chain Reaction (RT-qPCR)

TOS1 and HeLa cells (5 × 10^5^) were cultured in Roswell Park Memorial Institute (RPMI)-1640 (Nissui Pharmaceutical Co., Tokyo, Japan) and DMEM (Wako), respectively, at 37 °C in a 95% air/5% CO_2_ atmosphere for 24 h. Total RNA was extracted from the cells using a Direct-zol RNA MiniPrep Kit (Zymo Research, Irvine, CA, USA). cDNA was synthesized from the total RNA (1 µg) using a SuperScript VILO cDNA Synthesis Kit (Invitrogen, San Diego, CA, USA). qPCRs were run in a LightCycler 480 system using the LightCycler 480 Probes Master kit (Roche Diagnostics, Indianapolis, IN, USA). PCR primers for amplification of *VEGFR2* (forward: 5′-GAACATTTGGGAAATCTCTTGC-3′, reverse: 5′-CGGAAGAACAATGTAGTCTTTGC-3′), *PDGFRB* (forward: 5′-CATCTGCAAAACCACCATTG-3′, reverse: 5′-GAGACGTTGATGGATGACACC-3′), *KIT* (forward: 5′-TCAGCAAATGTCACAACAACC-3′, reverse: 5′-TCTCCATCGTTTACAAATACTGTAGTG-3′), *FLT3* (forward: 5′-TGGAATTTCTGGAATTTAAGTCG-3′, reverse: 5′-TTTCCCGTGGGTGACAAG-3′), *ABCG2* (forward: 5′-TGGCTTAGACTCAAGCACAGC-3′, reverse: 5′-TCGTCCCTGCTTAGACATCC-3′), and *GAPDH* (forward: 5′-AGCCACATCGCTCAGACAC-3′, reverse: 5′-GCCCAATACGACCAAATCC-3′) were designed using the Universal Probe Library Assay Design Center [https://www.roche-applied-science.com/sis/rtpcr/upl/acenter.jsp (accessed on 14 February 2019)]. Amplification products were separated on a 2.0% agarose gel. Bands were visualized by ethidium bromide staining.

### 2.11. Statistical Analysis 

Experimental results are presented as mean ± standard error (SE). Means were compared using the two-tailed Student’s *t*-test, and *p*-values < 0.05 were considered statistically significant. 

## 3. Results

### 3.1. Expression of Gb3 on RCC Cell Lines

There are various types of RCC cell lines [9,10] (Table 1). Our previous studies showed that the binding of SAL to Gb3 enhances the inhibition of cell proliferation and the effects of anticancer drugs [8,9].

Using flow cytometry and TLC, we first examined whether Gb3 is expressed on four RCC cell lines. As shown in Figure 1A,B, the surface expression of Gb3 was higher in TOS1 and TOS3 cells than in TOS3LN and ACHN cells. We previously reported that the effect of SAL depends on the expression level of Gb3 [6,8]. Therefore, we used TOS1 and TOS3 cells in subsequent experiments. Additionally, we confirmed the expression of VEGFR-2, FMS-like tyrosine kinase (FLT)-3, platelet-derived growth factor receptor (PDGFR)b, and the tyrosine-protein kinase kit (c-kit) gene on TOS1 cells (see Appendix A).

### 3.2. PI Uptake by and Viability of SAL-Treated TOS1 and TOS3 Cells

In a previous study, SAL treatment of Raji cells (Burkitt’s lymphoma cell line) expressing high levels of Gb3 at 37 °C increased the percentage of PI-positive cells with increasing treatment time, and the rate after 24 h of treatment was approximately the same as that after 30 min of treatment at 4 °C [7]. We previously reported that SAL does not reduce Raji cell viability even at concentrations of 100 µg/mL [7], and in HeLa cells, their morphology does not change until SAL concentrations of 200 µg/mL [8]. In the combined SAL and SU experiments with HeLa cells, the SAL concentration was 50 µg/mL. In this study, TOS1 and TOS3 treatments were performed using the same 50 µg/mL concentration of SAL. Additionally, RCC cell lines were treated with SAL at 4 °C for 30 min as a rapid method to determine the effect of SAL. To investigate the effect of SAL on PI uptake into RCC cells, we treated TOS1 and TOS3 cells with SAL (50 µg/mL) at 4 °C for 30 min, and then PI was added, and its uptake was observed by flow cytometry. The results showed that 41.5% of TOS1 cells and 21.1% of TOS3 cells treated with SAL were positive for PI (Figure 2A). SAL did not affect cell viability, even in the presence of PI, which normally induces apoptosis (Figure 2B). Based on these results, we decided to use TOS1, which showed a high expression of Gb3 and a high effect of SAL, in subsequent experiments.

### 3.3. Combined Effects of SAL and SU in TOS1 Cells

To confirm the cytotoxicity of SU on RCC cells, we treated TOS1 cells with SU alone and measured cell viability. The results showed that the viability of TOS1 cells decreased in a concentration- and time-dependent manner after SU treatment. At 48 h after treatment, the viability of TOS1 cells treated with 12.5 µM SU was similar to that of non-treated control cells; however, 25 μM SU moderately decreased cell viability to 70% (Figure 3A). Based on this result, we used 25 µM SU and 48 h treatment in subsequent combination treatment assays. In our previous study, SAL enhanced the cytotoxic effects of drugs [8,9]. Therefore, in this study, we observed whether the cytotoxic effect of SU on TOS1 cells was enhanced when combined with SAL pretreatment. The viability of TOS1 cells decreased to 70% after treatment with 25 μM SU alone and further decreased to 48% upon pretreatment with SAL (50 µg/mL) (Figure 3B). In addition, the number of annexin V-positive cells was significantly increased (*p* < 0.05) in cells pretreated with SAL as compared to cells treated with SU alone (Figure 3C). Additionally, microscopic images revealed cell atrophy, suggesting decreased viability (Figure 3D).

### 3.4. Effects of SAL on Intracellular Uptake and Extracellular Excretion of SU 

We investigated whether the decreased viability of TOS1 cells treated with a combination of SAL and SU was due to increased SU uptake by SAL treatment. As shown in Figure 4A, based on the time-series measurements of the intracellular fluorescence of SU, the uptake of 25 µM SU was significantly enhanced in SAL-treated TOS1 cells compared to non-SAL-treated control cells. We previously reported that the extracellular excretion of SU was delayed in SAL-treated HeLa cells [9]. However, such a delay was not observed in SAL-treated TOS1 cells, and no difference with the control cells was detected (Figure 4B).

### 3.5. Expression of ATP-Binding Cassette (ABC) Subfamily G (ABCG2) on TOS1 Cells

According to the results shown in Figure 4B, SU remained in the intracellular space of SAL-treated and non-treated TOS1 cells. SU is excreted by ABCG2 [11]. Therefore, we investigated whether ABCG2 is expressed on TOS1 cells. HeLa cells, which express ABCG2, were included as a positive control. RT-qPCR analysis confirmed the expression of *ABCG2* in HeLa cells; however, TOS1 cells did not express *ABCG2* (Figure 5).

### 3.6. Effect of SAL on Normal Human Renal Cells 

SAL enhanced the effect of SU on cancer cells; however, its effect on normal cells remained unknown. Therefore, we investigated the effect of SAL on HRPTEC. As shown in Figure 6A,B, although HRPTEC expressed Gb3, their expression level was lower than that of TOS1 cells. PI uptake assay results showed that SAL treatment did not increase PI uptake in HRPTEC (Figure 6C).

## 4. Discussion

In this study, we found that SAL recognizes and binds Gb3 on the surfaces of RCC cells, facilitating PI uptake. Although PI uptake is typically observed during the necrotic or late apoptotic phase of cell death [12], SAL enhanced PI uptake without inducing cell death. A similar observation was made in our previous studies using Raji and HeLa cells [8,9]. Moreover, the binding of SAL to Gb3 accelerated the uptake of the anti-RCC drug SU and consequently enhanced its cytotoxic activity. Based on this feature of increased cell surface permeability without the induction of apoptosis, we previously suggested that the combined use of doxorubicin and SAL in Raji cells and SU and SAL in HeLa cells increased intracellular drug concentrations and enhanced cytotoxic activity [7,8]. In the present study, we did not observe a delay in SU efflux, which can be explained by the fact that we did not observe ABCG2 expression in TOS1 cells. However, we found that SAL increased the intracellular SU concentration in TOS1 cells and decreased their viability. Importantly, SAL did not increase PI uptake in normal human renal cells. Herein, normal human renal cells were treated with SAL at a concentration of 50 µg/mL. We postulate that increasing the concentration of SAL did not enhance the effect of SAL on normal human renal cells because the amount of Gb3 in normal renal cells is lower than that in the RCC cell line. These results suggest that SAL can be used as a combination drug to enhance the efficacy of anti-RCC drugs. It is known that the therapeutic efficacy of two cancer drugs combined is higher than that of a single drug. Treatment of gastric cancer cells with the B subunit of Shiga toxin, which binds to Gb3 on the surfaces of gastric cancer cells, combined with SN38, the active metabolite of the topoisomerase inhibitor irinotecan, resulted in more than 100-fold greater cytotoxicity than treatment with irinotecan alone [13]. A combination of drugs that efficiently acts on the target cells without affecting normal cells is expected to allow drug dose reductions, thereby reducing side effects.

SU, which was used in combination with SAL in this study, has been indicated for metastatic RCC and exerts antitumor effects by inhibiting tyrosine kinases in signaling pathways, including the VEGFR-1–3, FLT-3, PDGFRb, and c-kit pathways. We confirmed that TOS1 cells express *VEGFR2*, *FLT3*, *PDGFRB,* and *KIT* (see Appendix A). These results indicated that SU might exert cytotoxic effects by inhibiting these tyrosine kinase receptors. SU exerts cytotoxicity in various solid tumors, including gastrointestinal stromal and pancreatic neuroendocrine tumors [14,15,16]. However, increased plasma levels of SU increase the incidence of side effects, and many patients are forced to discontinue treatment. We found that SU alone reduced the viability of TOS1 cells in a concentration- and time-dependent manner, whereas treatment with SAL alone did not affect cell viability. This suggests that SAL is effective in enhancing the cytotoxic activity of SU without affecting cell viability and, thus, may be a suitable candidate combination drug. Although other molecular-targeted agents (e.g., pazopanib, axitinib, everolimus) reduced the survival rate of TOS1 cells when used alone, not all of them were effective when combined with SAL (see Appendix A). Therefore, further studies are needed to determine which agents are effective in combination with SAL.

We observed that Gb3, a GSL, is expressed on the surfaces of TOS1 and TOS3 RCC cells and recognized and bound by SAL. The uptake of PI following SAL treatment was higher in TOS1 than in TOS3. This may be due to the higher Gb3 expression in TOS1 than in TOS3, which originate from human soft tissue with metastases and human kidney tissue with primary tumors, respectively. We thus speculated that the origin of the RCC cell line may be responsible for the differences in Gb3 expression levels. Although Gb3 is highly expressed in TOS1, glycosphingolipids with different mobilities, such as Gb4, are also expressed. In previous studies, we demonstrated that SAL binds strongly to Gb3 and not to other glycosphingolipids; Gb3 plays an essential role in exerting the effect of SAL [6,8]. However, functions other than those of Gb3 expressed in TOS1 remain to be elucidated. GSLs are abundant on cell surfaces and affect cell functions in various ways, including canceration, metastasis, and drug responses. Młynarczyk et al. showed that the content of Gb3 or GM3 in clear cell RCC is altered in a malignancy-grade-dependent manner [17]. Changes in the expression of GSLs on the cancer cell surface modulate genes involved in cell proliferation and apoptosis (i.e., oncogenesis), genes related to invasiveness and angiogenesis (i.e., promotion of metastasis), and drug transporter genes (i.e., drug resistance) [18]. GSLs are expressed characteristically in carcinomas, and Gb3 is particularly abundant on the surfaces of breast [19], rectal [20], testicular [21], bladder [22], lung [23], and gastric [14] cancer cells. Cancer cells expressing Gb3 show increased invasive and metastatic potential, as well as resistance to drugs, and several studies have evaluated the potential of Gb3 as a therapeutic marker or target. Tyler et al. reported that in cisplatin-resistant lung cancer, the expression of Gb3 and the drug excretion transporters MDR1 and MRP1 are upregulated, which facilitates the excretion of cisplatin from the cells, resulting in a decrease in drug concentration and the development of drug resistance [23]. Johansson et al. reported that treatment with verotoxin, which recognizes and binds Gb3, together with cisplatin, limited drug resistance to cisplatin and induced apoptosis of breast cancer cells [19]. Although the role of Gb3 in renal cancer remains to be elucidated, a combination of SAL, which recognizes and binds Gb3, and SU may be a new and useful therapeutic strategy.

Lectins, including SAL, have recently attracted molecular and cellular research attention as recipients of information from GSLs. Lectins are sugar-binding proteins widely distributed in the animal and plant kingdoms. Animal lectins have been suggested to be involved in infection, defense, cell differentiation, and cell adhesion [24]. Lectins are classified into several families based on the similarity of their carbohydrate recognition domains. Lectins widely distributed in fish eggs are characterized by their affinity for l-rhamnose and, thus, form the rhamnose-binding lectin (RBL) family. The RBL family includes not only SAL isolated from catfish (*Silurus asotus*) but also many other fish egg lectins, such as chum salmon (*Osmerus lanceoratus*, *O. keta*) lectins [25]. In addition to SAL, chum salmon egg lectin (CSL) 3 and *Crenomytilus grayanus* lectin (CGL) reportedly recognize Gb3 [26,27]. In contrast to SAL, which only reversibly increases membrane permeability when bound to Gb3, CSL3, and CGL induce cell death (i.e., apoptosis) when bound to Gb3 [26,27]. Further studies are needed to understand the changes in plasma membrane function that occur when SAL binds to Gb3 and the differential effects of lectins on Gb3-expressing cells.

## 5. Conclusions

SAL recognizes Gb3 expressed on the surfaces of RCC cells and increases the intracellular concentration of SU, a drug for RCC, enhancing its cytotoxicity to RCC cells. SAL may increase cell surface permeability and SU uptake by binding to Gb3. As SAL binds only to Gb3 and does not affect cell viability, sufficient cytotoxic activity may be achieved even when SU is administered at a sufficiently low dose not to cause side effects combined with SAL. However, there are many unknowns in the detailed molecular mechanism of SAL bound to Gb3, which requires further elucidation.

## Figures and Tables

**Figure 1 biomedicines-11-02317-f001:**
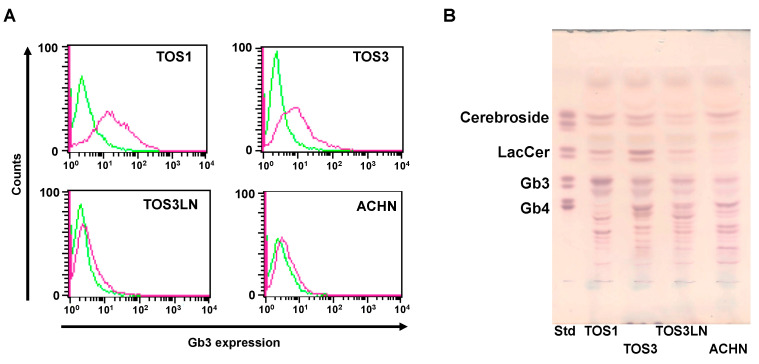
Comparison of Gb3 expression on RCC cell lines. (**A**) Flow-cytometric analysis of Gb3 expression on TOS1, TOS3, TOS3LN, and ACHN cells. Cells (2 × 10^5^) were treated with an anti-Gb3 mAb and an AF488-tagged goat anti-mouse mAb (red line). The level of Gb3 expression on TOS1, TOS3, TOS3LN, and ACHN cell surfaces was determined by flow cytometry. Control cells were treated with anti-Gb3 mAb alone. Fluorescence intensity of control cells: green line. (**B**) Total glycosphingolipids isolated from TOS1, TOS3, TOS3LN, and ACHN cells were separated by TLC using a solvent system as described in the Methods and were visualized by spraying orcinol-H_2_SO_4_ reagent. In the standard lane (Std), an aliquot of a standard mixture containing cerebrosides, LacCer, Gb3, and Gb4 was used.

**Figure 2 biomedicines-11-02317-f002:**
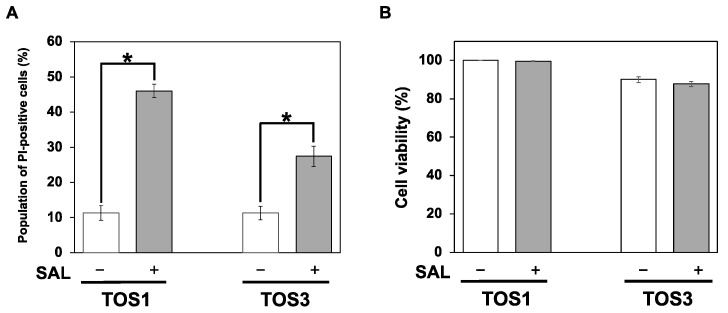
SAL enhances PI uptake but has no cytotoxic effect on TOS1 and TOS3 cells. (**A**) Cells (1 × 10^5^) were treated (+) or not (−) with SAL (50 µg/mL) at 37 °C for 24 h. The percentage of PI-positive cells was determined by flow cytometry. (**B**) Cells (5 × 10^3^) were treated (+) or not (−) with SAL (50 µg/mL) at 37 °C for 24 h. Cell viability was assessed using the trypan blue dye exclusion assay. Values represent mean values ± SEs of three independent experiments performed in triplicate. * *p* < 0.05 versus non-treated control cells.

**Figure 3 biomedicines-11-02317-f003:**
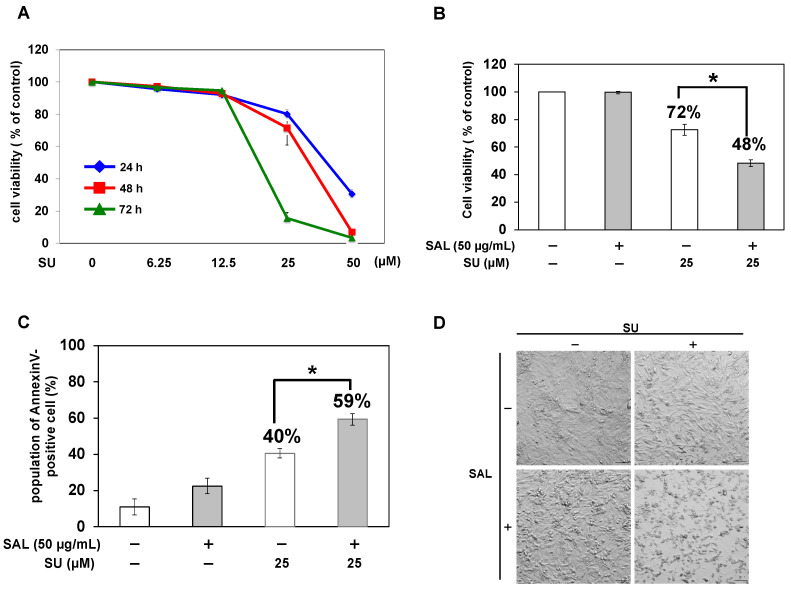
SAL enhances the antitumor effect of SU on TOS1 cells. (**A**) Cells (5 × 10^3^) were treated with SU (0, 6.25, 12.5, 25, and 50 µM) at 37 °C for 24, 48, and 72 h. (**B**) Cells (5 × 10^3^) were pretreated with SAL (0 and 50 µg/mL) at 37 °C for 24 h and then treated with SU (0 and 25 µM) at 37 °C for 48 h. Cell viability was measured using the WST-8 assay. Values represent mean values ± SEs of three independent experiments performed in triplicate. * *p* < 0.05 versus non-treated control cells. (**C**) Cells (5 × 10^3^) were pretreated with SAL (0 and 50 µg/mL) at 37 °C for 24 h and then treated with SU (0 and 25 µM) at 37 °C for 48 h. The percentage of annexin V-positive cells was determined by flow cytometry using a FACSCalibur™ Flow Cytometer. (**D**) Cells (5 × 10^3^) were pretreated (+) or not (−) with SAL (50 µg/mL) at 37 °C for 24 h and then treated with SU (0 and 25 µM) at 37 °C for 48 h. Bright-field microscopy images at a magnification of 10× are shown. Scale bar, 100 µm.

**Figure 4 biomedicines-11-02317-f004:**
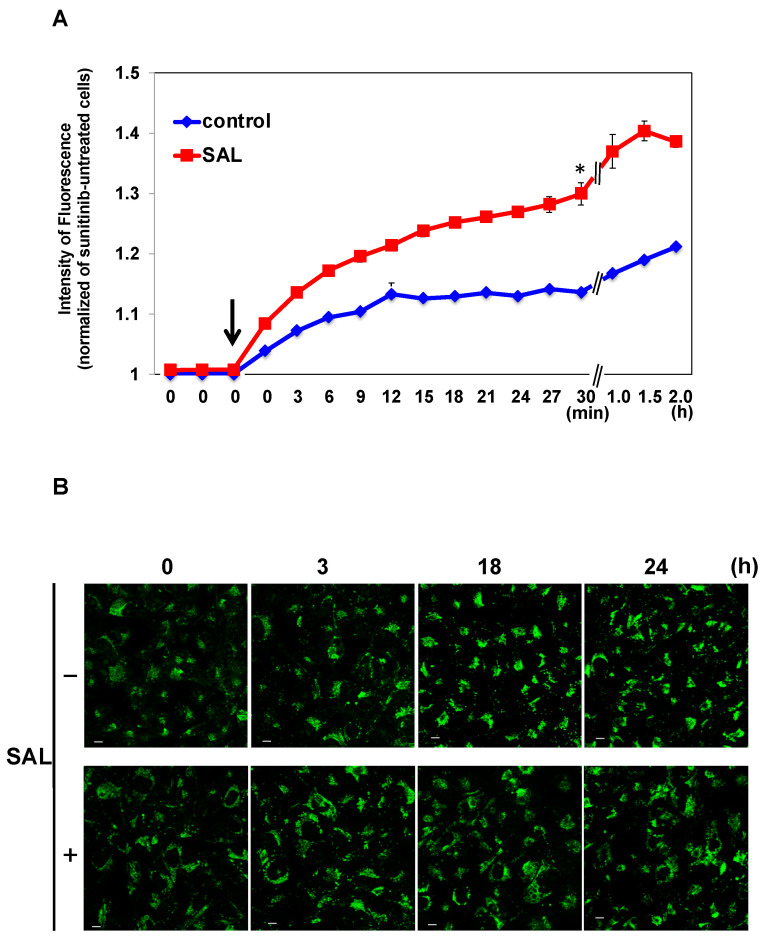
SAL promotes SU influx in TOS1 cells. (**A**) Time-course measurements of the autofluorescence of SU in cells using the Operetta CLS High Content Analysis System. TOS1 cells were treated (red squares) or not (blue rhombuses) with SAL (50 µg/mL) at 37 °C for 24 h. Subsequently, they were treated with SU (25 µM). SU influx was analyzed every 3 or 30 min after SU addition to the medium at time point 0. The arrow indicates the time immediately after SU addition. Values represent means ± SEs of three independent experiments performed in triplicate. * *p* < 0.05 versus non-treated control cells. (**B**) Cells (3 × 10^4^) were pretreated or not with SAL (50 µg/mL) at 37 °C for 24 h. Subsequently, the cells were treated with SU (12.5 µM) at 37 °C for 30 min. After SU was removed from the medium, the residual quantity in the cells was observed using confocal laser scanning microscopy at the indicated time points. A pseudo-cyan color represents SU. Photographs were captured using a 60× objective lens. Scale bar, 10 µm.

**Figure 5 biomedicines-11-02317-f005:**
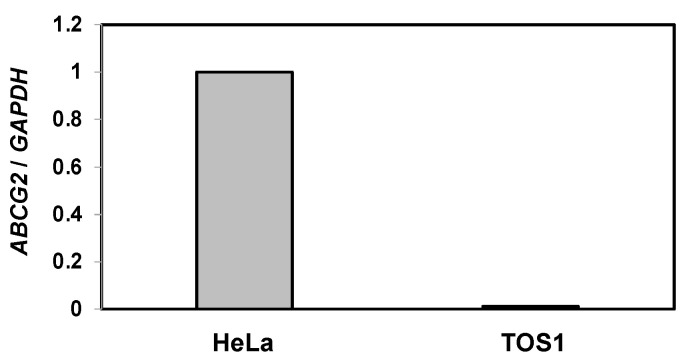
Expression of *ABCG2* in HeLa and TOS1 cells. Total RNA extracted from HeLa and TOS1 cells (5 × 10^4^) was analyzed by RT-qPCR using specific primers for *ABCG2* and *GAPDH* (control). Target gene expression was normalized to *GAPDH* expression, and the results are expressed as an n-fold increase over the control. Values represent means ± SEs of three independent triplicate experiments.

**Figure 6 biomedicines-11-02317-f006:**
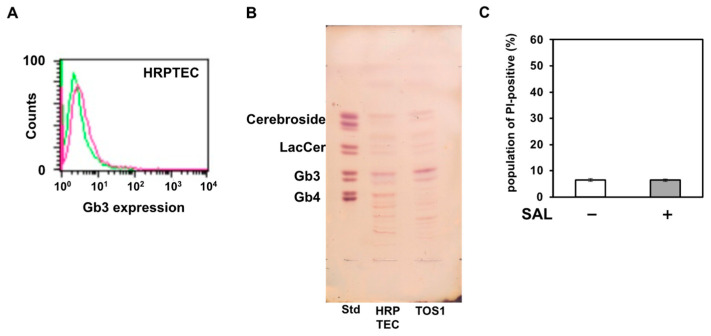
SAL does not affect normal human renal epithelial cells. (**A**) Flow-cytometric analysis of Gb3 on HRPTEC. Cells (2 × 10^5^) were treated with an anti-Gb3 mAb and an AF488-tagged goat anti-mouse mAb (red line). The level of Gb3 expression on surfaces of HRPTEC was determined by flow cytometry using a FACSCalibur™ Flow Cytometer. Fluorescence intensity of control cells: green line. (**B**) Total glycosphingolipids isolated from HRPTEC were separated by TLC using a solvent system as described in the Methods and visualized by spraying the orcinol-H_2_SO_4_ reagent. In the standard lane (Std), an aliquot of a standard mixture containing cerebrosides, lactosylceramide, Gb3, and Gb4 was used. (**C**) HRPTEC (1 × 10^5^) were treated (+) or not (−) with SAL (50 µg/mL) at 37 °C for 24 h. The population of PI-positive cells was determined by flow cytometry.

**Table 1 biomedicines-11-02317-t001:** Selected RCC cell lines currently in use.

Cell Line	Disease	Species	Source Organ		Sex	Established in
TOS1 *	Clear cell RCC	Human	Soft tissue	Metastasis	Male	1999
TOS2 *	Clear cell RCC	Human	Soft tissue	Metastasis	Male	1999
TOS3 *	Clear cell RCC	Human	Kidney	Primary	Male	1999
TOS3LN *	Clear cell RCC	Human	Lymph node	Metastasis	Male	1999
ACHN ^†^	Papillary RCC	Human	Pleural effusion	Metastasis	Male	1979
caki-1 ^†^	Clear cell RCC	Human	Skin	Metastasis	Male	1971
caki-2 ^†^	Clear cell RCC	Human	Kidney	Primary	Male	1971
786-O ^†^	Clear cell RCC	Human	Kidney	Primary	Male	1976
769-P ^†^	Clear cell RCC	Human	Kidney	Primary	Female	1976

More than 20 cell lines are widely used and stored in cell banks. Additionally, dozens of other cell lines have been established and used for research in selected laboratories. * Used in selected laboratories. ^†^ Available in cell banks.

## Data Availability

The datasets supporting the conclusions of this article are included within the article (and its additional files). They are not deposited in publicly available repositories. The datasets used and analyzed during the current study are available from the corresponding author upon reasonable request.

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
