# Peer review of "Catfish Egg Lectin Enhances the Cytotoxicity of Sunitinib on Gb3-Expressing Renal Cancer Cells"

_biomedicines, 2023, doi:10.3390/biomedicines11082317_

Round 1

Reviewer 1 Report

Overall, I think this is an entirely new outcomes, which could add new knowledge to this field and could pave a way for future therapeutic development against RCC.

 There are a few concerns that require the authors attention. These are listed below.

1.       Figure 1B:

In addition to Gb3, which is highly expressed in TOS-1 there are other glycosphingolipids including Gb4, and others which migrated slower than Gb4 or above Cerebroside. Please, commend on whether may play any roles (regulatory and/or functional).

2. Under citations/references: There are no 2022 and 2023 papers cited/referenced in this manuscript. The authors should make it a point to look for the relevant papers from 2022 and 2023 to include in the citations

Reviewer 2 Report

In this study, the authors evaluated whether the combined use of sunitinib (SU) and Silurus asotus egg lectin (SAL) would reduce the viability of be renal cell carcinoma (RCC) cells, and investigated whether the increased uptake and delayed excretion of therapeutic agents in RCC cells. The authors concluded that SAL recognized Gb3 expressed on the surfaces of RCC cells and increased cell surface permeability by binding to Gb3, which in turn, increased the intracellular concentration of SU and enhanced its cytotoxicity on RCC cells.

Comments

The reviewer has some concerns as follows:

1. The authors concluded that SAL enhanced the cytotoxicity of SU on Gb3-expressing RCC cells. However, all important experiments are only carried out on TOS1 cells, so whether the conclusions are only applicable to TOS1 cells and not other RCC cell lines? The authors should explain and clarify this issue.

2. TOS1 cells are from human soft tissue with metastasis and TOS3 cells are from human kidney tissue with primary. Are the cell lines from different sources the reason for the different effects of the test drugs on these two cell lines? The authors should explain and clarify this issue.

3. As shown in Figure 2, SAL enhances PI uptake but has no cytotoxic effect on TOS1 and TOS3 cells. However, there is only one concentration of SAL (50 µg/mL) used and cannot get the conclusion that SAL has no cytotoxic effects on TOS1 and TOS3 cells. This study is to test whether the combination of the two drugs will enhance the toxic effect on cancer cells, isn't it? It is necessary to find out the dosage that has the best cytotoxic effect to RCC cells when the two drugs are used together. The authors should explain and clarify this issue.

4. In Figure 6, the authors concluded that SAL did not affect normal human renal epithelial cells. However, there is only one concentration of SAL (50 µg/mL) used and this experimental design cannot draw this conclusion. The authors should explain and clarify this issue.

5. In Figure 4B, it needs to be confirmed at the dividing line between 18 and 24 hours.

6. In general, the presented results cannot support the conclusions.

Reviewer 3 Report

biomedicines-2556013

Title: Catfish egg lectin enhances the cytotoxicity of sunitinib on Gb3-expressing renal cancer cells

Authors: Jun Ito *, Shigeki Sugawara, Takeo Tatsuta, Masahiro Hosono, Makoto Sato

In general, the data in present studies are good and support the major conclusions of this manuscript. And due to two rounds of review, the manuscript has been significantly improved. However, following issues need to be considered prior to considering the manuscript of publication.

[Major concerns]

1.    Cell lines: The authors used four different types of RCC cell lines such as TOS1, TOS3, TOS3LN, and ACHN in their experiment. Among these, TOS1, TOS3, and TOS3LN are all clear cell renal cell carcinomas (RCCs) with only minor differences in their origins. If there are distinct genetic profiles among these cells, please provide further information. Additionally, please elaborate on the reasons why they respond differently to SAL.

2.    Abbreviations: The use of abbreviations when writing a paper has many advantages besides simplicity of expression. To use an abbreviation, first write the abbreviation in parentheses after the full name, and then use the abbreviation from Introduction to the final Conclusion. Abbreviations should only be used if they are repeatedly used and if they are not used again, only the full name should be used.

3.    Materials and Methods section - When naming a particular chemical company, you must provide location information such as company name, city and/or state (abbreviation in the USA and Canada) and country. Once you have named a company with the information, you should only mention a company’s name thereafter. Information about several companies is wrong, so check and correct it. It is generally well written in this paper, but there are a few mistakes, so find them and correct them. Examples: Information for Dojindo was written at Line 88. However, the information for Dojindo was written again at Line 139.

[Minor concerns]

1.    Figures 1, 2, and 5: TOS-1, TOS-3, and TOS-3LN should be written as TOS1, TOS3, and TOS3LN.

2.    Line 70: Please provide the full name of Dr. M. Sato to give him the full credit.

3.    Lines 77 and 78: Use just one form of company name: KURABO vs. Kurabo.

4.    Line 139: ‘Dojindo Laboratories’ is enough.

5.    Line 147: Provide the purchasing company for RPMI and DMEM.

6.    Line 174: Use RCC for ‘renal cell carcinoma’ at the title of Table 1.

7.    Line 276: HRPTEC had already been defined at Line 72. Therefore, just write it as HRPTEC.

8.    Line 355: MytiLec and CGL. In these two cases, it is appropriate to write the full name first and indicate the abbreviation in parentheses.

Overall, the manuscript can be considered to publication after major revision as indicated above.

biomedicines-2556013

Title: Catfish egg lectin enhances the cytotoxicity of sunitinib on Gb3-expressing renal cancer cells

Authors: Jun Ito *, Shigeki Sugawara, Takeo Tatsuta, Masahiro Hosono, Makoto Sato

In general, the data in present studies are good and support the major conclusions of this manuscript. And due to two rounds of review, the manuscript has been significantly improved. However, following issues need to be considered prior to considering the manuscript of publication.

[Major concerns]

1.    Cell lines: The authors used four different types of RCC cell lines such as TOS1, TOS3, TOS3LN, and ACHN in their experiment. Among these, TOS1, TOS3, and TOS3LN are all clear cell renal cell carcinomas (RCCs) with only minor differences in their origins. If there are distinct genetic profiles among these cells, please provide further information. Additionally, please elaborate on the reasons why they respond differently to SAL.

2.    Abbreviations: The use of abbreviations when writing a paper has many advantages besides simplicity of expression. To use an abbreviation, first write the abbreviation in parentheses after the full name, and then use the abbreviation from Introduction to the final Conclusion. Abbreviations should only be used if they are repeatedly used and if they are not used again, only the full name should be used.

3.    Materials and Methods section - When naming a particular chemical company, you must provide location information such as company name, city and/or state (abbreviation in the USA and Canada) and country. Once you have named a company with the information, you should only mention a company’s name thereafter. Information about several companies is wrong, so check and correct it. It is generally well written in this paper, but there are a few mistakes, so find them and correct them. Examples: Information for Dojindo was written at Line 88. However, the information for Dojindo was written again at Line 139.

[Minor concerns]

1.    Figures 1, 2, and 5: TOS-1, TOS-3, and TOS-3LN should be written as TOS1, TOS3, and TOS3LN.

2.    Line 70: Please provide the full name of Dr. M. Sato to give him the full credit.

3.    Lines 77 and 78: Use just one form of company name: KURABO vs. Kurabo.

4.    Line 139: ‘Dojindo Laboratories’ is enough.

5.    Line 147: Provide the purchasing company for RPMI and DMEM.

6.    Line 174: Use RCC for ‘renal cell carcinoma’ at the title of Table 1.

7.    Line 276: HRPTEC had already been defined at Line 72. Therefore, just write it as HRPTEC.

8.    Line 355: MytiLec and CGL. In these two cases, it is appropriate to write the full name first and indicate the abbreviation in parentheses.

Overall, the manuscript can be considered to publication after major revision as indicated above.

Round 2

Reviewer 2 Report

This revised manuscript can be accepted. No further comments.

Reviewer 3 Report

I recommend considering the adoption of the resubmited paper, as all the issues highlighted during the first review process have been appropriately addressed and corrected.